# HPV Vaccination Adherence in Working-Age Men: A Systematic Review and Meta-Analysis

**DOI:** 10.3390/vaccines11020443

**Published:** 2023-02-15

**Authors:** Carlotta Amantea, Nazario Foschi, Filippo Gavi, Ivan Borrelli, Maria Francesca Rossi, Valerio Spuntarelli, Pierluigi Russo, Maria Rosaria Gualano, Paolo Emilio Santoro, Umberto Moscato

**Affiliations:** 1Postgraduate School of Occupational Medicine, Università Cattolica del Sacro Cuore, Largo Francesco Vito 1, 00168 Rome, Italy; 2Department of Urology, Fondazione Policlinico Universitario Agostino Gemelli, Largo Francesco Vito 1, 00168 Rome, Italy; 3Postgraduate School of Urology, Università Cattolica del Sacro Cuore, Largo Francesco Vito 1, 00168 Rome, Italy; 4Department of Health Science and Public Health, Università Cattolica del Sacro Cuore, Largo Francesco Vito 1, 00168 Rome, Italy; 5School of Medicine, Saint Camillus International University of Health Sciences, UniCamillus, 00131 Rome, Italy; 6Department of Women, Children and Public Health Sciences, Fondazione Policlinico Universitario Agostino Gemelli IRCCS, Largo Francesco Vito 1, 00168 Rome, Italy

**Keywords:** HPV vaccine, young workers, adherence, male, systematic review

## Abstract

Background: Human papillomavirus (HPV) infection is the most common sexually transmitted viral infection in the world. HPV vaccination adherence rates in men are generally lower than in women. The aim of this systematic review and meta-analysis was to assess adherence to HPV vaccination in young working-age males (18–30 years old). Methods: A systematic review was performed using three databases: PubMed, Scopus, and Web of Science, according to the Preferred Reporting Items for Systematic Reviews and Meta-Analyses (PRISMA). Results: After duplicate removal, the initial search resulted in 478 eligible papers. With the exclusion of 425 papers after screening the abstracts, full texts of 53 articles were reviewed. Subsequently, 45 were excluded. Among the eight studies included, four (50%) examined the vaccination adherence in young adults through data registered in nationwide insurance or private companies’ databases, three (37.5%) in young adults in different settings through data collected from surveys and questionnaires, and one (12.5%) an HPV vaccination campaign in a family medicine residency practice. Conclusion: Adherence to HPV vaccination in men of working age (18–30 years) does not appear to be adequate (pooled prevalence 11%). In order to achieve a higher level of compliance, it is important to place an emphasis on vaccination campaigns in schools as well as in the workplace, after consultation with and approval from local, regional, and federal public health agencies.

## 1. Introduction

Human papillomavirus (HPV) infection is the most common sexually transmitted viral infection in the world [1]. Every year, HPV causes 630,000 cancer cases in men and women (more than 95% of cervical cancer is due to HPV infection), posing a serious threat to global public health [2]. HPV has a huge cellular specificity for epithelial tissues and a great degree of adaptation, a feature that has led it to change over time. They are a group of viruses capable of producing subclinical infections of long evolution with low viral replication and clinical infections, managing to evade the defense mechanisms of the host. Ultimately, they participate in the carcinogenesis of cutaneous, oral, and anogenital tumors [3]. Nowadays, more than 200 subtypes of HPV have been identified, and about 85 of these have been recognized in the human body [4]. Among these, 45 subtypes are responsible for infection of the genital tract, while the others could lead to skin disease [5]. The infection is transmitted more by direct contact, even if the skin lesions can be transmitted indirectly, through contaminated surfaces [6]. In addition, microtraumas expose the basal layer keratinocytes and facilitate contagion [3]. Genital HPV lesion is the most prevalent sexually transmitted infection (STI) in the world [7]. It is estimated that 75% of sexually active adults will have at least one HPV infection in their lifetime. The most common genotypes are 6 and 11, present in about 80% of oral lesions and more than 90% of genital lesions [7]. The risk of infection is low in the absence of sexual relations, but it grows rapidly with increases in sexual partners. Other risk factors include age at first sexual intercourse, immunosuppression, and the presence of other STIs, such as herpes simplex. The infection can be transmitted directly with any form of sexual contact, even without penetration. The incidence of anal infection is high among men who have sex with men (MSM) [8]. In men, infection appears to be less persistent than in women [3]. The points where infection is easily transmitted are: transition epithelium in the uterine cervix and anal region, epithelia of the salivary glands of the oral cavity, and tonsillar crypts of the oropharynx, hair follicles, and eccrine and apocrine glands, in addition to the epidermis [3]. The entry of the virus at the level of the epithelium allows for the binding of the same with the heparan sulfate proteoglycans of the basal cells and of the basement membrane that then acts as receptor. The virus binds through its L1 protein, allowing the internalization of the capsid within the host cell. Virus protein L2 instead mediates entry of the viral genome into the nucleus. Infection starts with the transcription of the viral genome [9]. The viral replication process is divided into three stages: (1) DNA amplification, in which E1 and E2 proteins play a significant role; (2) viral replication in growing infected cells; and (3) genome amplification and virus creation. Maintenance might run for months or years [10]. HPV ends its life cycle in epithelial cells remaining hidden through the phenomena of evasion of the innate immune response and delaying the adaptive immune response. This characteristic is responsible for the persistent infection of HPV and its role in the carcinogenesis process [3]. Usually, keratinocytes and Langerhans cells (antigen-presenting cells present in the squamous epithelium) can initiate a response to viral pathogens, but E6 and E7 proteins interfere in this mechanism. This way the specific antigen immune response does not occur [10]. In the case of a clinically evident infection, there is a local immune response involving CD8+ cytotoxic T lymphocytes and T helper 1 CD4+, which produce interleukin 2 (IL-2) and interferon gamma (IFN-gamma) and recognize the viral protein E6, E7, and E2. Complete elimination of the virus may not occur, leading to its latency. Its reactivation is often observed in immunosuppressed patients or in HIV patients [11]. HPV infection and the immune response are different in men and women; in fact, in women there is a peak of prevalence in the late teens and twenties, declining steadily throughout the subsequent decades [12], while men acquire infection in the late teens and the prevalence remains constant throughout the subsequent decades [13,14]. About 70% of women seroconvert after cervical HPV infection with antibodies to the major coat protein L1 [15], in contrast to men where only about 30% do [16]. However, despite the low antibody response in natural infections, men develop important humoral immune responses to VLP vaccines with virtually 100% seroconversion. This explains the importance of vaccination in men, which is still not widespread in some countries despite extensive screening programs for women and girls.

The HPV vaccine is strongly advised for regular immunization at ages 11 or 12, according to the Center for Disease Control and Prevention (CDC). (Vaccinations can begin as early as age 9). If you weren’t sufficiently immunized when you were younger, the CDC also advises immunization for everyone up until age 26. Depending on the age at first immunization, a series of two or three doses of the HPV vaccine is administered (under 15 years of age, two doses are required to complete vaccination cycles, and over 15 years of age, three doses are required). If they were not sufficiently immunized when they were younger, some individuals between the ages of 27 and 45 may elect to receive the HPV vaccination after consulting with their physician. Because more persons in this age group have previously been exposed to HPV, HPV vaccination for those in this age range offers less benefit [17]. Vaccination campaigns are an essential prevention strategy in public health. Nowadays, immunization schedules from childhood to adulthood are organized around the world to address the problem of vaccine-preventable diseases. Vaccination in the workplace is one of the protection tools that the employer can adopt as a measure to prevent vaccine-preventable diseases. Real world data are necessary for the organization of vaccination and prevention campaigns that can be effective and have an impact on the health of workers. Regarding HPV vaccination, however, adherence in young males remains low. According to the report of the Italian Ministry of Health (www.salute.gov.it accessed on 10 February 2023) as of 31st December 2021, in males born in 1997, 1998, and 1999, vaccination coverage with a complete cycle was 0.49%, 0.62%, and 0.90%, respectively. In order to adapt effective prevention measures to the increase in adherence to the HPV vaccine, data on vaccination coverage in young workers are essential, since these are the years in which the administration of the HPV vaccine has important health benefits.

A recent systematic review identified several factors influencing HPV vaccination in men. The highest rates of vaccination tended to be associated with the following identified factors: men aged 10 to 20 years, adolescents with single parents, adolescents with single parents, higher parental knowledge and awareness of HPV vaccination, low economic status, individuals with public insurance, males living in urban areas, individuals receiving other vaccinations, frequent visits to clinics, and receiving health care providers’ recommendations [18]. Working age is the age range in which an individual is considered fit and available to enter the world of work. This age is established by national laws and may vary from country to country. In many countries, the minimum working age is 16 or 18, but in some countries it may be lower. Working age is also used to describe the age range in which an individual is considered to be at their maximum earning potential and able to participate actively in the labor market. This is generally considered to be between the ages of 25 and 54, although it can vary based on factors such as education level and health status. Therefore, we conducted our systematic review on young workers.

The aim of this systematic review and meta-analysis was to assess adherence to HPV vaccination in young working-age males (18–30 years old). This paper is the third of a series of literature reviews regarding public health and urology that our research group has completed. [19,20]

## 2. Materials and Methods

A systematic review was performed using three databases: PubMed, Scopus, and Web of Science, according to the Preferred Reporting Items for Systematic Reviews and Meta-Analyses (PRISMA) [21]. Primary studies written in English were examined. No time limits were applied. The search query was structured according to the PICO methodology and contained keywords that recognized young men of working age between 18 and 30 years (P, population) who had received HPV vaccination (I, intervention) and who had adhered fully or partially to the immunization program (O, outcome); a control (C) was not included in the review.

To do the bibliographic search, the following query was employed:

((HPV OR “Human papilloma virus” OR “Human papillomavirus” OR “Papilloma virus” OR “wart virus” OR “alpha papillomavirus” OR papillomaviridae)

AND

(“HPV Vaccine” OR “Human Papilloma Virus Vaccine” OR “wart virus vaccine” OR vaccin*)

AND

(male OR boy* OR man OR men OR guy* OR “working age population” OR work OR employ*)

AND

(adherence OR adhesion OR compliance))

All retrieved papers were reviewed to find those that indicated associations between HPV vaccination adherence and young men of working age. After retrieving the articles from all the selected databases, duplicates were removed, and the initial screening by title and abstract was performed using the website tool Rayyan [22], which permitted independent screening of the articles by the researchers according to the triple-blind methodology to reduce selection bias.

### 2.1. Inclusion Criteria

Articles addressing HPV vaccination in young adults of working age were considered for title and abstract screening. To screen full-text articles, stricter inclusion criteria were applied: the article had to mention adherence to HPV vaccination, comprehended as a partial (1 doses) or full (3–2 doses) vaccination cycle; it had to include the working-age between 18–30 years and male population. Primary studies written in English were examined. No time limits or country restrictions were applied to our search.

### 2.2. Exclusion Criteria

Our review excluded studies that did not include HPV vaccination adherence among male workers between 18–30 years old. Exclusion criteria also included the lack of full-text articles or article types that were incompatible with the aims of the research (systematic reviews, reviews, conference proceedings, commentaries, etc.).

### 2.3. Data Extraction and Synthesis

Two researchers (F.G. and C.A.) extracted data from the included papers and presented their findings in an Excel spreadsheet. The first author (F.G.) extracted data from the included research, while the second author (C.A.) validated the extracted data. If consensus could not be achieved, a third author (M.F.R.) was set to step in. The information to be collected was predetermined and comprised the following: author; year and country where the research was done; study design; sample size; age; type of HPV vaccine administered; percentage of young working-age boys completely vaccinated for HPV (3–2 doses) or vaccinated with incomplete cycle (one doses).

### 2.4. Quality Assessment

The Newcastle–Ottawa Scale was used to evaluate the methodological quality of each of the studies included [23].

### 2.5. Statistical Analysis

The pooled prevalence of HPV vaccines coverage in young males of working age was estimated using a meta-analysis. The I2 statistic was used to assess study heterogeneity. The percentages of 25%, 50%, and 75% were considered as low, moderate, and high heterogeneity, respectively. A random-effects model for meta-analysis was applied. Publication bias was assessed with the Egger regression model. A *p* value of 0.05 was used as the significance threshold. A forest plot was used to illustrate the overall predominance. *p* < 0.05 was used as the significance threshold. All statistical analyses were performed using version 14 of the STATA software (Stata Corp, College Station, TX, USA).

## 3. Results

The initial search resulted in 709 relevant studies among all three databases (Pubmed, Web of Science and Scopus). After duplicate removal (231 duplicates articles), the initial search resulted in 478 eligible papers. With the exclusion of 425 papers after screening the abstracts, full texts of 53 articles were reviewed. Subsequently 45 were excluded. Figure 1 shows the complete article selection process. All eight studies [24,25,26,27,28,29,30,31] were cohort studies and had a low risk of bias assessed by the Newcastle–Ottawa Scale, and all were assessed to have at least adequate methodological quality (score ≥ 6). Seven were retrospective studies [24,25,26,27,28,30,31], and one was a cross sectional study [29]. Eight studies were included for qualitative synthesis (Table 1). Seven studies were included for quantitative synthesis. Among the eight studies, four (50%) examined the vaccination adherence in young adults through data registered in nationwide, insurance, or private companies’ databases [27,28,29,30], three (37.5%) in young adults in different settings through data collected from surveys and questionnaires [24,25,26], one (12.5%) an HPV vaccination campaign in a family medicine residency practice [31]. All the studies were conducted in the United States of America (USA). The total number of patients considered was 147,465. Hubbard et al. [30] analyzed data of 134,867 (90%). One study reported data of 6267 (4.2%) patients [24], one study reported data of 3591 (2.4%) patients [25], one study had a sample size of 1849 (2%) patients [29], and four studies had a sample of less than 600 (1%) patients [24,27,28,29].

Srivastav et al. [24] analyzed combined data from 2013–2015 National Health Interview Survey, a nationally representative probability-based health survey of the non-institutionalized U.S. population ≥ 18 years for vaccination coverage. Among 101,091 patients, 6267 (9.5%) received at least 1 vaccine for HPV. This study does not differentiate between participants vaccinated with one, two, or three doses; therefore, it was not possible to determinate how many participants received full vaccination against HPV. Due to this, the study was excluded from the quantitative synthesis (meta-analysis).

Boakye et al. [25] analyzed data from patients of 18–26 years of age collected from 2014 to 2015 by the National Health Interview Survey (U.S.A) (*n* = 7588). A total of 3591 (49.8%) were males, and only 144 (4%) completed HPV vaccination. Participants were less likely to initiate the vaccine if they were men or 0.19 (0.16–0.23). Less likely to start and finish the HPV vaccination were men, those with only a high school degree or less education, and people born outside of the USA Poor HPV vaccine compliance was observed. The likelihood of beginning or finishing the HPV vaccine was improved by seeing a healthcare provider. In this study, 26.8% of participants had initiated HPV vaccination, with a much higher adhesion rate in women than men (42.0% women and 11.4% men, *p* < 0.0001); women also showcased a higher completion rate than men, with 27.1% completing the vaccination. Partial vaccination was higher in United States citizens, with 28.6% US-born participants receiving at least one dose of the vaccine, against 14.5% of foreign participants starting the vaccination cycle. Furthermore, 16.8% of participants born in the United States completed the vaccination cycle, which was completed in only 7.6% of foreign participants. Education was highlighted as an important factor in HPC vaccination uptake: participants with a high school degree or a lower education level had 54% lower odds of initiating the vaccination process and 70% lower odds of completing the vaccination cycle, compared to those with a college degree or a higher level of education (aOR: 0.46; 95% CI: 0.32–0.64 and aOR: 0.30; 95% CI: 0.19–0.47). Finally, participants who had visited the doctor’s office at least one time in the year prior to the study had greater odds of initiating and completing the vaccination process.

Thomas et al. [26] recruited 400 male patients (13–26 years) at a hospital-based teen health center and a health department sexually transmitted disease (STD) clinic between 2013 and 2015 for an epidemiologic study on HPV. A total of 46 (14.5%) patients (19–26 years) reported to be fully vaccinated for HPV. Overall, only 26% of patients (13–26) have received at least one vaccination for HPV. However, the data for those who were partially vaccinated are not reported exclusively for the 19–26-year-old population Thomas et al. highlighted important differences in HPV vaccination uptake based on age range: 69.9% of participants aged between 14 and 18 years initiated the vaccination cycle by receiving at least one dose, while this rate was only 32.1% of participants between 19 and 21 years of age, and 4.9% in participants between 22 and 26 years.

Keim-Malpass et al. [27] conducted a retrospective study based on data of the University of Virginia’s Clinical Data Repository (CDR) on patients vaccinated for HPV between 2009 to 2013. Overall, 3371 were included. There were 75 male patients (19–26 years) included, and only 6 (8%) of them received full HPV vaccination. Interestingly, investigating vaccination through a repository, the authors were able to perform the Wald test of time trend, which highlighted a significant (*p* < 0.0001) decrease in HPV vaccination completion from 2009 to 2013. The Wald test of time trend also highlighted that, while there were no male participants in the years prior to 2010, a progressive increase in male participants was observed from 2011 to 2013, with 22%, 47%, and 45% of people, respectively, initiating the vaccination cycle for HPV being males.

Hirth et al. [28] analyzed data from one commercial insurance company. Among 118 males (19–26 years), 18 (15.2%) completed the HPV vaccination. The authors highlighted that older males were less likely to complete the vaccination cycled compared to younger males aged 9 to 12. Furthermore, male participants who initiated the series in 2009 were three times more likely to complete it compared to males who initiated in 2006. Finally, the authors highlighted that, in male participants who received only two doses of the HPV vaccination, the mean time interval between the first and second dose was more than twice that of the interval for participants who completed the cycle; in participants who received all three doses, but completed it in more than 365 days, the interval between the first and second injection was more than 2.5 times higher than that of completers, and the mean interval between second and third dose was 288 days for participants who completed the vaccinations in more than 365 days.

Wiener et al. [29] conducted a cross-sectional study from the 2015 Behavioral Risk Factory Surveillance Survey (BRFSS) database. Among 1849 male patients (18–29 years), 112 (6%) received all three doses of the HPV vaccines. Concerning partial vaccination, 9.3% of male participants aged between 18 and 29 years of age received one or two vaccination doses, but did not complete the HPV vaccination, while the vast majority (86.5%) never started the vaccination process.

Hubbard et al. [30] used the data collected in the Marketscan Commercial Claims and Encounters (CCE) database from 2011 to 2017. A total of 134,867 male patients (18–27 years) were included in the analysis. A total of 8777 (6.8%) patients completed the HPV vaccination. The authors also highlighted that male participants from 9 to 12 years of age were less likely to adhere to the recommended schedule than male participants between 18 and 27 years (RR: 0.92; 95% CI: 0.90–0.95) amongst those who completed the vaccine cycle. However, male participants between 18 and 27 years were the least likely to complete the vaccination doses, with higher completion rates in males aged 13–17 (50.6%) and 9–12 (37.6%).

McGaffey et al. [31] reported the results of a vaccination campaign in a residency outpatient office located in a disadvantaged urban neighborhood in Pittsburgh, Pennsylvania, from 31 October 2015 to 31 October 2016. Among young adults (18–26 years), the increase in the HPV completion rate was 11.3% in males and 9.5% in females, but no significant statistical differences were found by gender (*p* > 0.05). Out of 326 males (18–26 years), 40 (30.1%) completed HPV vaccinations pre intervention. Out of 381 males (18–26 years), 55 (41.4%) completed HPV vaccinations (*p* < 0.002) post intervention. The authors highlighted an important increase in HPV vaccination adhesion and completion after the intervention in young adult males ages 18 to 26 years, with an increase of 6 percentage points in vaccination initiation for males (*p* < 0.015), and an increase of 11.3 percentage points in vaccination completion (*p* < 0.002), showcasing the importance of information campaigns regarding vaccine-preventable diseases.

Seven studies were included in the quantitative analysis of HPV vaccination adherence in working-age men. The total number of patients included was 140,950. All studies had a low risk of bias assessed by the Newcastle–Ottawa Scale, and all were assessed to have at least an adequate methodological quality (score ≥ 6). Heterogeneity was I^2^ = 96.72% *p* < 0.001, and no significant publication bias was found (Egger test = 1.89 95% CI = −19.77–23.37, *p* = 0.83). A random effects model was used due to the high heterogeneity. The pooled HPV vaccination coverage was 11% (95% CI: 8–14%).

## 4. Discussion

Among sexually transmitted diseases (STDs), HPV infection is considered the most common worldwide [7,32]. Persistent HPV infection is associated with more than 5% of all cancers in the world [33] and more than half of all malignancies in the world [34]. One of the main problems of HPV as an oncovirus is a striking gap between the times of diagnosis of the chronic infection and its early stages [35]. The early detection of HPV infection and HPV-induced lesions is crucial for cancer prevention.

The World Health Organization (WHO) has recommended HPV vaccination as a routine immunization for girls and female people [36], and a lot of countries have increased national HPV vaccination programs for women [37]. Fortunately, during the last several years, HPV vaccination programs have been including all genders as it seems to be more effective for successfully acquiring herd immunity against the virus [37,38]. However, few countries have included young males in their national HPV vaccination programs [39,40]. Loke et al. reported a very low HPV vaccination rates among males (1.1–31.7%) compared to females (2.4–94.4%) in the United States and Canada [41]. This systematic review and meta-analysis show a low adherence (overall 11%) of HPV vaccination in young males of working age (18–30 years). HPV vaccination adherence rates in men are generally lower than in women, and they also vary by age group. For boys aged 13–17, vaccination rates are around 20–30%. For men aged 18–26, vaccination rates are around 10–15%. For men aged 27–45, vaccination rates are around 5–10%. For girls aged 13–17, vaccination rates are around 60–70% [42]. For women aged 18–26, vaccination rates are around 30–40%. For women aged 27–45, vaccination rates are around 20–25% [42]. It is important to note that these are approximate estimates, and that the adherence rate varies between countries and regions. In addition, the rates included are for the first dose of the vaccine, and the adherence rate for the completion of the vaccine series may be even lower. The results of our review also show higher adherence to starting the vaccine cycle (partial vaccination) and not completing it (full vaccination), as shown in Table 1. It is worth noting that HPV vaccination for boys and men has been less emphasized than for girls and women, and many healthcare providers do not routinely recommend the vaccine for boys and men. Additionally, there is often less awareness among men and their parents about the fact that the HPV vaccine can also protect against several HPV-related cancers in men. Up to 90% of HPV-related cancers may be avoided via vaccination. Vaccination protects males against developing anogenital condylomas and other malignancies associated with infections, including those of the penis, anus, and base of the tongue. When both sexes are vaccinated, the spread of the virus is reduced [43]. However, there is a lack of access to the vaccine for men, as it is not covered by many insurance plans. All these factors contribute to lower HPV vaccination adherence in men compared to women, and data from our meta-analysis confirm this trend. The major problem concerns the lack of knowledge of the prevention of pathologies associated with HPV [44]. Greater education on this issue allows for greater adherence to vaccination campaigns [45]. Those who are not recommended for vaccination by the general practitioners are less likely to get HPV vaccination compared to young people who were recommended [46]. Vaccine coverage of men, not just women, has been included in nationwide school-based vaccination programs across the globe, but uptake is irregular, knowledge is deficient, and misconceptions remain [47].

It is important to understand the factors affecting HPV vaccine initiation. Previous reviews showed race and ethnicity, age, previous vaccination history, health insurance status, personal knowledge and awareness of HPV, and parental knowledge and education levels are significant predictors for HPV vaccination uptake [48].

Religious exemptions are among the factors that impact HPV vaccine adherence. In the United States, for instance, each state determines its own religious exemptions to vaccination, and numerous states have significantly enlarged the list. Because the HPV vaccination protects against a sexually transmitted disease, it is often listed among the immunizations for which students may obtain religious exemptions [49].

The Middle East is another example of vaccination refusal owing to religious and/or cultural restrictions. In research by Ortashi O. et al., the authors note that fewer than half of male students in the United Arab Emirates would accept HPV vaccination, while around 30% were undecided. A total of 25 percent of the reasons deemed most likely to deter pupils from taking the vaccination were opposition from a religious authority [50].

Health care provider and general practitioners’ recommendations are some of the most significant factors leading to the initiation of HPV vaccination [51,52,53]. However, despite positive evidence, the delivery of recommendations associated with HPV vaccine in men is a challenge that must be overcome. Some reviews explain a gap linked to recommendations according to sex that originated from the provider’s lack of knowledge or opinions [54,55]. Thus, health associations must induce consistent recommendations for immunization from the health care providers through training for health professionals to enhance the rates of male HPV vaccination [25,56,57].

In addition, male people who had taken other vaccinations (eg, tetanus, diphtheria, and pertussis; meningitis; influenza; and hepatitis B) had higher HPV vaccination rates. There is the potential to develop a program to link the initiation of HPV vaccination to other vaccinations conducted at the same age [40]. Parental awareness of HPV vaccination is an essential facilitator of HPV vaccinations in children, because the lack of knowledge regarding the HPV vaccine is one of the causes of low HPV vaccination rates in men [25,58,59]. To encourage rates of HPV vaccine uptake in men, national policies should include parents as decision makers and aim to increase their knowledge and awareness [40]. Vaccine hesitancy is one of the greatest health challenges facing the world today [60,61]. This is a complicated issue due to a lack of authority trust, scientific illiteracy, and biased media coverage. The COVID-19 pandemic led to significant problems in HPV vaccination. Although preventive health services have resumed, capacity for in-person preventive care remains lower than pre-pandemic levels [47]. To date, studies of COVID-19 vaccines have shown no particular interactions or interferences with other vaccinations. Even a document from the U.S. Centers for Disease Control and Prevention (CDC) [62] reveals no warnings against using this vaccination in conjunction with others. Current recommendations in Italy suggest waiting two weeks between COVID-19 immunization and other vaccines for increased confidence of effectiveness. It seems to be appropriate to provide at least a two-week waiting period between the SARS-CoV-2 vaccination and the anti-HPV vaccination. Regarding human papillomavirus, much research that has been published in academic journals suggests a connection between infertility and HPV infection. In HPV-infected patients, a reduction in sperm motility, viability, and morphology has been reported. This is in comparison to people who are not infected with HPV. Luttmer et al. (2016) discovered that people with infertility had a high frequency of HPV infection, with HPV16 (high-risk) being the most prevalent genotype observed in their research [63,64]. Based on our results, the only study with an adherence rate higher than 15% is from McGaffey et. al. [31], who conducted an HPV vaccination campaign increasing the vaccination adherence rate dramatically. The authors demonstrated a significant increase in HPV vaccination adherence and completion after the intervention (the information campaign) in young adult males aged 18–26 years. There was a 6% increase in vaccination initiation for males and an 11.3% increase in vaccination completion. This highlights the significance of information campaigns on vaccine-preventable diseases. HPV vaccination campaigns in general help to reduce health disparities. In some countries, access to healthcare and vaccines is limited, leading to higher rates of HPV-related illnesses. By implementing HPV vaccination campaigns, we can help to reduce health disparities and improve overall public health [65]. Furthermore, HPV vaccination can also help to control the spread of HPV and reduce the overall burden of the virus on healthcare systems [66]. Additionally, vaccinating against HPV can also save cost in the long run, as HPV-related cancer treatments are often costly and require significant resources from healthcare systems [65]. School-based HPV vaccination programs allow for higher vaccine coverage and contribute to sociodemographic equality [67,68,69,70]. A good solution to increase the global uptake is to offer catch-up vaccinations, preferably in the school health system [67,68,69,70] at youth health clinics and in health facilities at universities, as reported by an Australian research study, after consultation with and approval from local, regional, and federal public health agencies. There should be provision for HPV vaccination education campaigns to be adopted in the workplace as well, especially for young employees. Vaccination is not only very effective but also extremely safe [71]. According to the WHO, it is also appropriate to vaccinate young males. In addition, by offering vaccinations to boys and young men whose parents previously declined the vaccine and by preventing future HPV-related cancer among high-risk groups, individuals would reduce health disparities and empower young men to make informed decisions regarding the prevention of future HPV-related diseases. The promotion of HPV vaccination requires conscientious public health measures. Understanding the cost components connected with vaccination delivery techniques and their efficacy can assist in identifying potential for expansion. Economic costs frame a bigger picture on the total monetary and non-monetary implications of implementing HPV vaccination strategies [72]. Statewide implementation of quality improvements visits, centralized reminder and recall, and school-located vaccination were each cost-effective at increasing HPV vaccine coverage, relative to no intervention [73]. Genital HPV infection is very common among men, with an ongoing international study estimating a prevalence of 65.2% in asymptomatic males aged 18–70 years [74]. The problem of the asymptomatic nature of the infection among the male population contributes to a greater spread of the virus. Greater awareness on this issue and a greater study of the natural history of the disease in the general population could help ensure greater therapeutic adherence to vaccination [75].

Regarding the type of recommended vaccine, the results of our review show that the 4-valent and 9-valent vaccine are the most widely used. In fact, HPV 9-valent vaccine (directed to protect against HPV types 6, 11, 16, 18, 31, 33, 45, 52, and 58) and the quadrivalent HPV vaccine (directed against HPV types 6, 11, 16, and 18) are the more common, although there is also the bivalent vaccine (directed against HPV types 16 and 18). The 9-valent HPV vaccine is similar in composition to the quadrivalent vaccine, using virus-like particles to elicit immune responses [76], but it included more HPV types [77]. Available evidence has demonstrated the efficacy and safety of the 9-valent HPV vaccine [77,78,79] also among male people [80] The vaccine has the potential to prevent most cervical neoplasia and reduce the incidences of other HPV-associated cancers in males and females. The HPV vaccination has a reassuring safety record; nonetheless, like other medications, it may have side effects, such as the case of acute disseminated encephalomyelitis after its administration in a 14-year-old boy [81]. Clearly, the benefits of vaccination outweigh the risks, but the vaccinating physician should always evaluate the patients’ medical history and family history before the administration. Considering the cost for each dose, the 9-valent vaccine could prove to be more cost-effective than the current quadrivalent vaccine in male and female-only vaccination programs [82,83]. With these factors in mind, family physicians should recommend to patients the 9-valent vaccine over the current quadrivalent vaccine. Another interesting aspect of the study is that all the included studies were carried out in the United States of America (USA). The USA represents the world’s most important economic power [84]. It is the nation with the greatest total health expenditure, at 15.3 percent of GDP, over twice the OECD average of $6714 per capita per year [85]. Despite this, HPV vaccine adherence seems to be quite poor in young working-age men (11% overall) (Figure 2). In impoverished nations, there are concerns about adherence, but more importantly, about the availability of HPV vaccination and vaccines in general. With the support of Gavi, the Vaccine Alliance (Gavi), the United Nations Children’s Fund (Unicef), and the World Health Organization (WHO) will administer the HPV vaccine through schools of Sierra Leone, targeting 153,991 10-year-old girls who will receive two doses each over a six-month period.

This systematic review and meta-analysis have some limitations. We have not found any eligible studies reporting data on HPV vaccination rates in workers in other countries other than the USA, which could lead to a selection bias. Only articles written in English were included, which limited the number of articles that could be evaluated for the qualitative analysis and the data that could be included in the quantitative analysis. Additionally, the meta-analysis carried out in the current research has included few studies. To provide a more accurate picture of the HPV vaccination adherence in men worldwide, future studies should consider extending vaccine research to more ethnically varied groups.

## 5. Conclusions

Adherence to HPV vaccination in males does not appear to be adequate. This systematic review investigates a population, young men of working age (18–30 years), in whom there is evidence of extremely poor adherence to vaccinations (overall 11%). In order to achieve a higher level of compliance, it is important to place an emphasis on vaccination campaigns and the dissemination of information about the risks of Human Papilloma Virus infection as well as the benefits of vaccination. After consultation with and approval from local, regional, and federal public health agencies, this information could be provided in schools as well as in the workplace.

## Figures and Tables

**Figure 1 vaccines-11-00443-f001:**
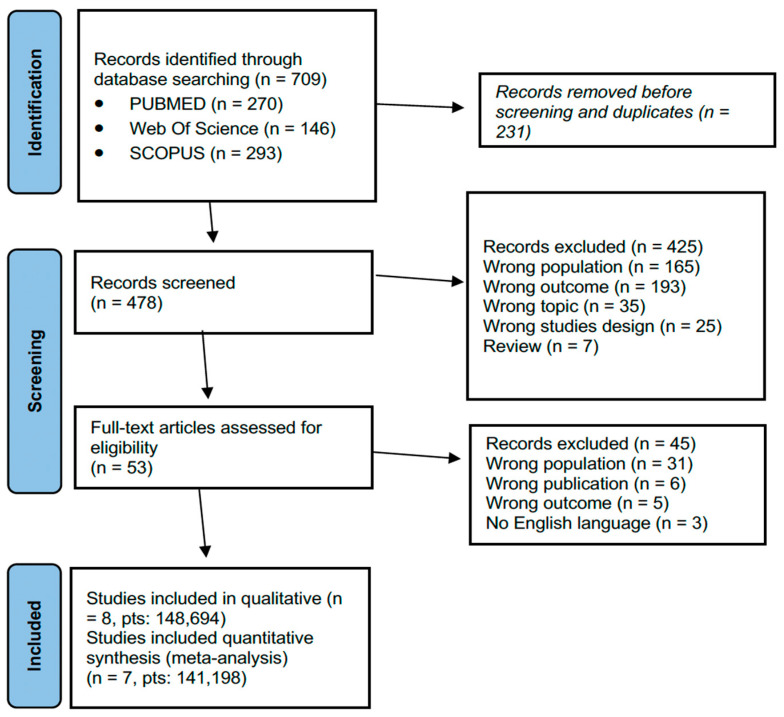
PRISMA flowchart for study selection.

**Figure 2 vaccines-11-00443-f002:**
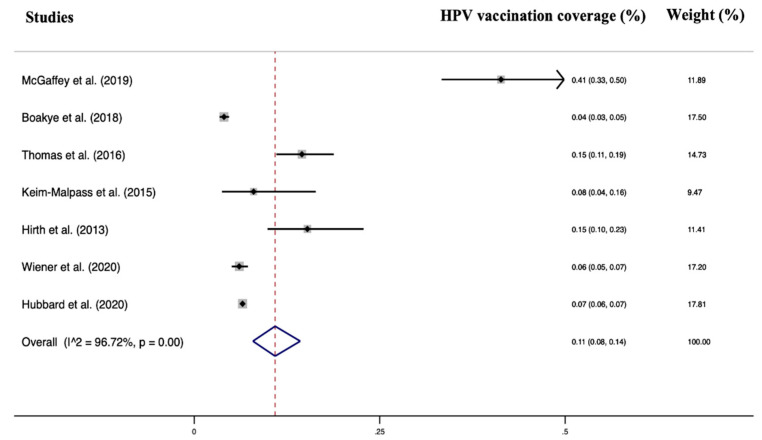
Forest plot of the included studies [23,24,25,26,27,28,29,30].

**Table 1 vaccines-11-00443-t001:** Main characteristic of the included studies.

Author	Year	Country	Study Design	Vaccine Type	Participants	Fully Vaccinated ** *n* (%)	Partially Vaccinated *** *n* (%)
Srivastav et al. [24]	2019	USA	Retrospective study	NR *	6267	NR *	NR *
Boakye et al. [25]	2018	USA	Retrospective study	NR *	3591	144 (4.0)	410 (11.4)
Thomas et al. [26]	2016	USA	Retrospective study	NR *	317	46 (14.5)	NR *
Keim-Malpass et al. [27]	2015	USA	Retrospective study	NR *	75	6 (8.0)	69 (92)
Hirth et al. [28]	2013	USA	Retrospective study	Quadrivalent HPV vaccine	118	18 (15.0)	101 (85)
Wiener et al. [29]	2020	USA	Cross-sectional study	NR *	1849	111 (6.0)	159 (8.6)
Hubbard et al. [30]	2020	USA	Retrospective study	Gardasil 4/Gardasil 9	134,867	16,184 (6.8)	126,101 (93.5)
McGaffey et al. [31]	2019	USA	Retrospective study	Gardasil 4/Gardasil 9	133	55 (41.4)	NR *

* NR: not reported; ** Fully vaccinated: 3–2 doses; *** Partially vaccinated: one dose.

## Data Availability

Data are available upon reasonable request.

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
