# Peer review of "HPV Vaccination Adherence in Working-Age Men: A Systematic Review and Meta-Analysis"

_vaccines, 2023, doi:10.3390/vaccines11020443_

Round 1

Reviewer 1 Report

This manuscript provides data about HPV vaccination in working-age men.  The authors observed a weak adherence to HPV vaccination recommendations in men of working age (18-30 years). This topic is appropriate for this journal.   A few comments for improvement are listed below.

1.     Introduction.  The Introduction is very long and probably provides more information about the classification of all papillomaviruses in the world than is necessary for this meta-analysis.  Suggest that the last line on page 1 and the top 24 lines of page 2 be deleted.  Re-start with the sentence about nowadays more than 200 subtypes.

2.     Discussion, page 9, line 16 from top, sentence beginning with it is important to understand the factors affecting HPV vaccine initiation.  The authors should specifically mention religious exemptions.  In the USA, each state decides upon its religious exemptions to vaccination.  Several states have greatly expanded their list of religious exemptions.  Because HPV protects against a sexually transmitted disease, HPV is often included among the vaccines eligible for religious exemption to students.  At this time in the USA, there will be few if any changes to this exemption policy by the states.  In fact, the number of exemptions are increasing, not decreasing.

Here is a new reference to cite: R. Pierik, On religious and secular exemptions:  A case study of childhood vaccination waivers. Ethnicities17:220, 2017

3.     Another example of vaccine refusal because of religious and/or cultural barriers in the Middle East.  See article by O. Ortashi et al, Acceptability of HPV among made university students in the United Arab Emirates.  VACCINE 31: 5141, 2013.  The authors point out that less than half of male students in the U.A.E would accept HPV Vaccination.  Add this information into the article and cite the reference.

4.     Reference citation 58 in the Discussion, page 9, 19 lines from bottom.  The sentence describes reasons for vaccine hesitancy.  Reference 58 is from 2008.  Also add a newer reference about vaccine residency written during the years of the COVID-19 epidemic, 2020-2022.

5.     Discussion, 22 lines from top of page 10; sentence about a good solution to increase uptake in the school health system (references 66-69).  References 66,67 and older (years 2011 to 2012).  Only reference 69 is new, from 2021.  Furthermore, Reference 69 was a study from Australia, not the USA.  Further, the authors of reference 69 state that their study is the first study to demonstrate that HPV vaccination of boys in a school setting may have a positive effect.  Please add this information into text that this is not a USA study.

Comment to authors. There is enormous vaccine hesitancy in the USA since the COVID-19 epidemic, with reluctance to introduce any new immunization programs into public schools in the USA.  The U.S Congress just passed legislation (signed by the President) which states the soldiers in the army no longer need to accept COVID-19 vaccination.  Since the U.S Congress has decreed that they do not want mandatory COVID-19 vaccination on young men in the U.S.A, it seems unlikely that any school system in the U.S.A is going to encourage HPV vaccination of male adolescents in school.

6.     Three lines from bottom of page 10.  Sentence about safety of HPV vaccine in boys and reference 79.  Reference 79 is from 2015.  It is true that the HPV vaccines have an excellent safety record.  Nevertheless, a bothersome case report about a severe adverse event was published in 2021 by a reputable neurology group in a reputable neurology journal.  This case should be briefly mentioned and cited.  See article by A. Wellnitz, et al.  Fatal acute hemorrhagic leukoencephalitis following immunization against HPV in a 14-year-old-boy. Child Neurology Open, May 2021. PMID: 34046515.

7.     Last sentence of the Conclusion. At least in the USA, there would be opposition to providing HPV information in many public schools. Suggest that the authors add the following phrase in front of the last sentence: After consultation with and approval from local, regional and federal public health agencies, this information could be provided in schools as well as in the workplace.

8.     Last sentence of Abstract. See Comment 7. Suggest that the authors advise consultation with local, regional and federal public health agencies before an emphasis is placed on vaccination campaigns on adolescent boys in schools. 

Author Response

Dear reviewer,

Thank you for your comments. We listed the answers below:

  1. We edited the introduction as you suggested.
  2. Thanking the reviewer for the suggestion, we have added in the text of the paper in review mode a comment about this and the recommended reference.
  3. Thanking the reviewer for the suggestion, we have added in the text of the paper in review mode a comment about this and the recommended reference.
  4. Thanking the reviewer, we added the references: [Chido-Amajuoyi OG, Pande M, Agbajogu C, Yu RK, Cunningham S, Shete S. HPV Vaccination Uptake, Hesitancy, and Refusal: Observations of Health-Care Professionals During the COVID-19 Pandemic. JNCI Cancer Spectr. 2022 Jul 1;6(4):pkac053. doi: 10.1093/jncics/pkac053. PMID: 35900184; PMCID: PMC9382715.]

  5. Thanking the reviewer for the comment, we have edited the text in review mode and specified that the reference [69] refers to an Australian study.
  6. Thanking the reviewer for the suggestion, we have added in the text of the paper in review mode a comment about this and the recommended reference.

  7. As suggested by the reviewer we integrated and added as the last sentence of the conclusions "After consultation with and approval from local, regional and federal public health agencies, this information could be provided in schools as well as in the workplace”.

  8. As suggested by the reviewer we added as the last sentence of the abstract "after consultation with and approval from local, regional and federal public health agencies."

Reviewer 2 Report

This review is very well written and detailed, I just have some minor comments of understanding. Congratulations to the authors for their work.

Author Response

Dear reviewer,

Thank you very much for your precious comments. We listed the answers below:

  1. We added the references as suggested.
  2. Thank you for your suggestion. We made the decision to include only articles written in English language to maintain a higher standard while performing the articles search and screening.
  3. Thank you for your comment. Although this article does not report the absolute prevalence of full vaccinated  or partially vaccinated patients, it provides the prevalence of patients vaccinated with at least 1 dose. The data is significant, only 9.5% of the total. We couldn’t include it in the meta analysis but it was still worth mentioning it.

  4. We added the limits of the study in the discussion.